# Open-source low-cost cardiac optical mapping system

**Dmitry Rybashlykov[1], Jaclyn Brennan[2], Zexu Lin[2], Igor R. Efimov[2]\*, Roman Syunyaev[1,3]\***

**1** Human Physiology Lab, Moscow Institute of Physics and Technology, Moscow, Russia, **2** Department of Biomedical Engineering, The George Washington University, Washington DC, United States of America, **3** Sechenov University, Moscow, Russia

\* irefimov@gmail.com (IRE); roman.syunyaev@gmail.com (RS)

**Data Availability Statement:** All optical mapping recordings files is available from the zenodo database DOI:10.5281/zenodo.5557829

**Funding:** The research was supported by Russian Foundation for Basic Research (https://www.rfbr.

## Abstract

Fluorescent imaging with voltage- or calcium-sensitive dyes, known as optical mapping, is one of the indispensable modern techniques to study cardiac or neural electrophysiology, unsurpassed by temporal and spatial resolution. High-speed CMOS cameras capable of optical registration of action potential propagation are in general very costly. We present a complete solution priced below US$1,000 (including camera and lens) at the moment of publication with an open-source image acquisition and processing software. We demonstrate that the iDS UI-3130CP rev.2 camera we used in this study is capable of 200x200 977 frames per second (FPS) action potential recordings from rodent hearts, with the signal-to-noise-ratio of a conditioned signal of 16 ± 10. A comparison with a specialized MiCAM Ultimate-L camera has shown that signal-to-noise ratio (SNR) while lower is sufficient for accurate measurements of AP waveform, conduction velocity (± 0.04 m/s) and action potential duration (± 7ms) in mouse and rat hearts. We used 4-aminopyridine to prolong the action potential duration in mouse heart, thus demonstrating that the proposed system is adequate for pharmacological studies.

## Introduction

An optical technique of measurements of cellular transmembrane voltage *via* potentiometric dyes was introduced in the 1970s, known today as optical mapping [1–4]. Potentiometric dye molecules bind to cell membranes and undergo either molecular motion or an electronic redistribution upon excitation and emission [5]. The changes of the external electrical field affect transition energy, corresponding emission spectrum can be detected and recorded. Further advances in the field include calcium-sensitive dyes (changing emission spectrum upon binding with calcium ions) [6], metabolic imaging (*via* intrinsic NADH and/or FAD fluorescence) [7], simultaneous mapping of voltage and calcium [8,9], simultaneous imaging from the several sides of the heart (panoramic mapping) [10,11], and transmural imaging via long wavelength dyes [12].

One notable advantage of optical mapping in comparison to traditional multielectrode arrays is high spatial and temporal resolution that makes it possible to accurately track the

ru/rffi/eng) grant 19-29-04111 (to RS) and Leducq Foundation (https://www.fondationleducq.org/) project RHYTHM (to IE). The funders had no role in study design, data collection and analysis, decision to publish, or preparation of the manuscript.

**Competing interests:** NO authors have competing interests.

rapidly propagating excitation wavefronts in ventricular and atrial arrhythmias [13]. However high spatio-temporal resolution requires highly specialized cameras: 100x100 pixels, 1,000 frames per second (FPS) and digital image acquisition hardware. It typically has a high price of US$50,000–100,000. This price is prohibitively high for education and some research applications, for example, several cameras are required for multiparametric and panoramic optical mapping. Recently, high speed USB 3.0 machine vision-specialized industrial CMOS cameras entered the mass market eliminating the need for specialized data-acquisition systems and, thus, reducing the price of a fast imaging system. Previously, Lee et al. [14] have demonstrated that it is possible to optically map pig and rabbit hearts with relatively inexpensive ($600–1200) CMOS cameras, as it was possible with inexpensive CCD cameras [15]. In particular, Lee et al. have shown that action potential (AP) recordings up to 1,000 Hz and SNR of up to ~50 (defined as (AP Amplitude)/(SD during diastolic intervals)) are possible in large animal hearts with USB3.0 iDS (Imaging Development Systems, GmbH) cameras. Unfortunately, this method was not applied to rodent models, which are much more popular models compared to pigs and rabbits, and not made available to the wider research community.

In our open-source research and development presented here we used iDS UI-3130CP-M-GL (~$700US including lens) and iDS Software Suite programming interface that makes it possible to customize image acquisition (Fig 1). Here, we demonstrate capabilities of action potential recordings in the two most popular laboratory animal models in cardiovascular research: rat and mouse hearts. Small rodent hearts are more challenging for optical recordings as compared to large animal hearts due to much lower optical signal intensity. We compared this inexpensive solution to the state-of-the-art MiCAM ULTIMA-L system, which has superior SNR (Fig 2), but at a much higher price approaching $100,000. New inexpensive optical mapping system provides sufficient quality data to track activation and repolarization sequences and action potential duration (APD).

## Materials and methods

### iDS UI-3130CP camera and software

In this study we used a UI-3130CP camera from Imaging Development Systems, which is capable of 10-bit recordings at resolutions up to 800 by 600 pixels. The "uEye cockpit" image acquisition software provided by iDS has neither an option to save recordings in lossless format, nor an option to make 10-bit recordings. Therefore, we have developed a custom open-source image acquisition application using the C++ API provided by iDS (https://github.com/humanphysiologylab/ueyemappingWin).

The custom software was designed to make recordings with a resolution of 200x200 pixels and a framerate of 977 frames per second (FPS). The high frequency recordings were possible because of the reduction of the active part of the sensor to the smaller area in the center. Although the framerate up to 1400 FPS is possible at 120x120 pixels resolution, this resulted in a narrow (5.5 degrees) field of view. Since direct capture to solid-state drive resulted in frame loss and uneven FPS, software was designed to capture the recording directly to the RAM and to transfer it to the storage after the recording has finished. This approach requires RAM size large enough to store the whole recording. For example, a 5 second recording at 200x200 resolution at 977 FPS requires 375 MB of memory. Considering this data memory requirement, at least 2 GB of RAM is recommended. The software interface allows users to visualize signal intensity and change the recording gain and frame rate before the image acquisition (S1 Fig), which is essential to adjust LED intensity when signal intensity is too low or saturated. The file format of the data is covered in the software user manual. Binary iDS camera recordings are

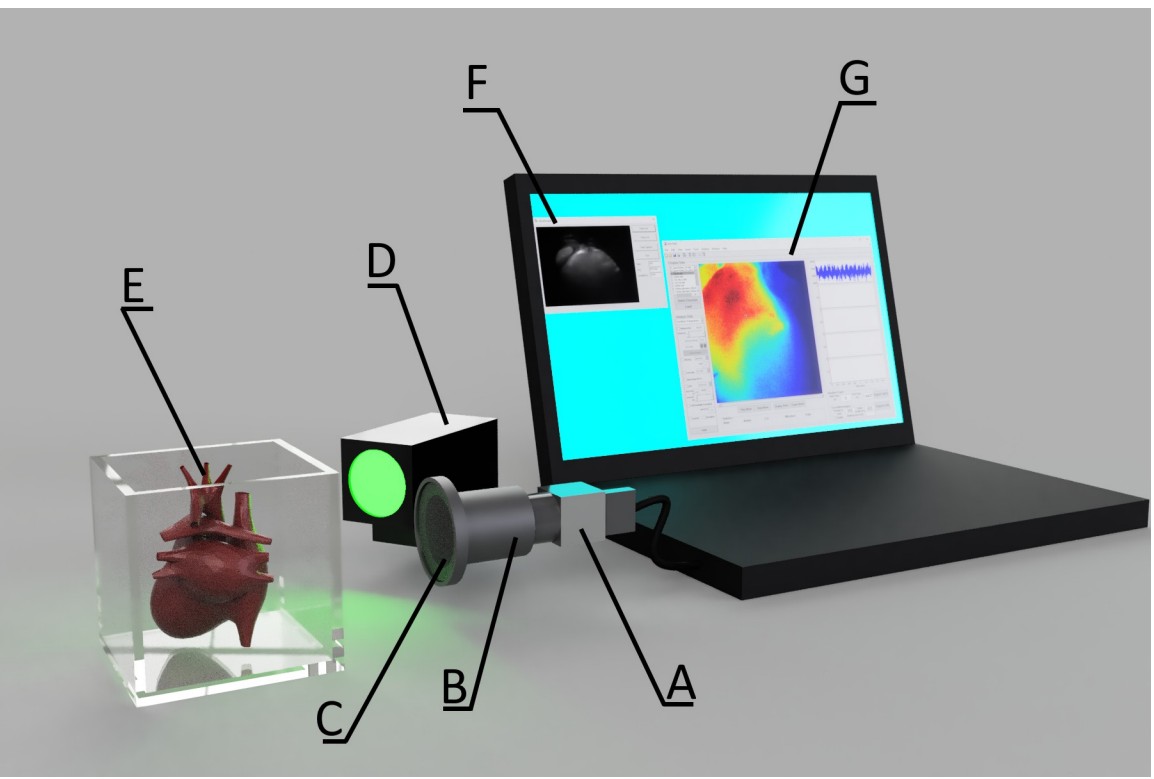

**Fig 1. The experimental setup for the iDS camera.** (A) iDS UI-3130CP M.GL camera (977Hz sampling frequency). (B) Pentax C60607KP lens. (C) 650nm long-pass filter. (D) Green excitation LED (530nm wavelength). (E) Perfusion chamber with heart. (F) **iDS** Recorder application. (G) Open source RHYTHM1.2 software, based on Matlab and C++.

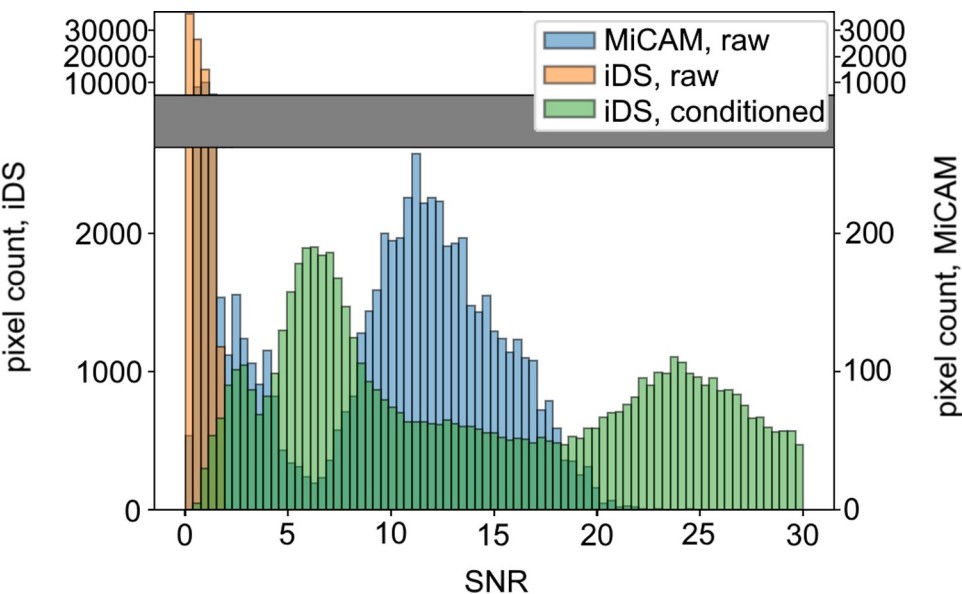

**Fig 2. Histogram of SNR in recordings from mouse ventricles (n = 6).** Orange represents SNR of the raw signal obtained by iDS, green represents SNR from the same pixels after signal conditioning, blue represents corresponding SNR of the raw signal obtained by MiCAM.

compatible with popular open-source RHYTHM signal processing and analysis software [9], which we update regularly.

## Optical mapping protocol

All experimental animal protocols were approved by the Institutional Animal Care and Use Committee at The George Washington University and conform to the NIH Guide for the Care and Usage of Laboratory animals.

We followed the protocol described earlier [16]. Briefly, animals were anesthetized via isoflurane vapors, after ensuring thoracotomy heart was excised and quickly cannulated. Hearts were Langendorff-perfused with Tyrode's solution (in mM: 128.2 NaCl, 4.7 KCl, 1.05 MgCl2, 1.3 CaCl2, 1.19 NaH2PO4, 20 NaHCO3, 11.1 Glucose), electromechanically uncoupled with 5–10 μM blebbistatin and stained with voltage-sensitive dye Di-4-ANEPPS. Left ventricle was paced at 80–150 ms pacing cycle length (PCL), PCL of 150 ms was fast enough to suppress sinus rhythm in all experiments. The dye was excited with 520 nm LED (Prizmatix UHP-Mic-LED-520), and fluorescent emission was captured through a long-pass filter (650nm) (Fig 1). Resulting optical signal was recorded sequentially by either the iDS camera or the MiCAM from the same camera position: after recording several sequences at varying pacing cycle lengths (PCL), it was removed, and the iDS camera was installed in its place (Fig 1).

## Pharmacological protocol

We used a transient outward current ($I_{to}$) blocker 4-Aminopyridine (4-AP) [17–24] (Millipore Sigma, Cat. 278575). 250 mM 4-AP stock solution was prepared with pH adjusted to 7.4 using 1 M hydrochloric acid (Fisher Scientific, SA48-1). Small quantities of the 4-AP stock solution were added to the modified Tyrode's perfusion solution and perfused for 10 min to reach a final working concentration of 5.6 mM.

## Data conditioning and analysis

Signal conditioning included ensemble averaging both in space (gaussian filter with window of 3 by 3 pixels for MiCAM and 5 by 5 pixels for iDS) and in time (all beats were averaged in a 2 s recording). The signals after all intermediate steps of signal conditioning for both cameras are shown in the S2 Fig. Raw iDS-recorded signal was, in general, noisier as compared to signals from MiCAM camera. While spatial filtering reduced the SNR to a level adequate to measure upstroke for activation sequence and conduction velocity (CV) analysis, both spatial and temporal averaging were required to measure APD. Therefore, only the former step of signal conditioning was used for the activation sequence and conduction velocity (CV) measurements. Conduction velocity was calculated using RHYTHM 1.2 [9]. with the algorithm earlier described by Bayly et al [25]. Signal-to-noise ratio (SNR) was calculated as the ratio of the root mean square amplitude to the root mean square noise, where the amplitude of noise is evaluated at resting potential.

APD was measured at 80% repolarization (APD80). The noise amplitude affected apparent resting potential in the recordings, hampering the comparison of APD80 measured by two cameras. Therefore, prior to APD calculation, resting potential level for each pixel was determined by gaussian filter ($\sigma$ = 7ms, truncated at $4^*\sigma$). Noisy and oversaturated areas were excluded semi-automatically by choosing appropriate SNR and signal intensity cutoff levels. APD outlier boundaries were calculated as Q1 - 1.5*IQR and Q3 + 1.5*IQR, where Q1 and Q3 denote first and third quartile respectively and IQR = Q3 - Q1 (interquartile range). Any value exceeding those boundaries was considered an outlier and was excluded from statistical analysis.

Signal processing and analysis were done with Rhythm 1.2 software [9], while APD and SNR comparison between two cameras were done as described above with custom python scripts.

## Results

The capabilities and limitations of the iDS camera system were tested in comparison with the more expensive state-of-the-art MiCAM Ultimate-L system on Langendorff-perfused mouse and rat hearts. We used the traditional optical mapping setup [16] shown on Fig 1 (see "Materials and Methods" for details on signal acquisition and processing). Raw iDS signal recordings were, in general, quite noisy: SNR was 0.5 ± 0.4 for mouse hearts (Fig 2; here and below standard errors are reported). Signal processing (binning and ensemble averaging, see Methods for details) increased SNR to 16 ± 10 for the mouse heart (Fig 2, processed signal SNR and signal intensity maps are shown in S3 Fig). The comparison of representative iDS and MiCAM optical mapping systems recordings are shown in 3C–3H and Fig 4C–4H. The processed signal clearly reproduces both depolarization phase and general AP waveform.

In order to verify the accuracy of the conditioned signal waveform we have compared the mouse APD measured by the two cameras in control (n = 6) and during administration of Ito blocker 4-aminopyridine (5.6 mM, n = 6). The comparison at 150 ms PCL is summarized in Fig 5. APD80 measurements in left ventricle by two cameras differed by 7 ± 12 ms in control and 5 ± 5 ms in 4-AP; in right ventricle: 5 ± 9 ms in control, 6 ± 5 ms in 4-AP. The difference is partially due to the fact that recordings were not simultaneous and the field of view was slightly different. However, we found no statistically significant differences between APD measured by the cameras neither in left nor in right ventricle (Fig 5). It was possible to measure drug effect by the cheaper system: the iDS camera system registered 19 ± 6 ms and 20 ± 6 ms action potential prolongation by 4-AP in left ventricles (p = 0.0035, paired t-test, Fig 5A) and right ventricles (p = 0.0035, paired t-test, Fig 5B), respectively, which corresponds to low concentration 4-AP measurements in isolated cell experiments [17].

The APD restitution is depicted in Fig 6A. We did observe the reduction of APD with decreasing PCL with both cameras: over a change of PCL from 150 ms to 80 ms mean APD has decreased by 19 ms and 18 ms in left and right ventricles respectively according to iDS and by 24 ms in both ventricles according to MiCAM, corresponding to relatively shallow slope of the restitution of the mouse heart [26,27].

Figs 7 and 8 show comparisons of activation maps (A, C) and APD maps (B, D) measured in mouse and rat hearts by iDS and MiCAM. Since the depolarization phase is less prone to noise, activation sequence and, consequently, CV could be determined more accurately than APD with the iDS camera. Root mean square deviation (RMSD) between measurements by two cameras was *4 cm/s* for longitudinal CV and 2 cm/s for transversal CV (Fig 6B). The CV restitution measurements with iDS camera were consistent with previous studies [28,29] with a slight reduction in CV when pacing frequency was increased (or PCL reduced). For example, at PCL = 150 ms longitudinal CV was 51 ± 4 cm/s, transversal CV was 29 ± 3 cm/s; while at PCL = 90 ms longitudinal CV was 47 ± 3 cm/s, transversal CV was 25 ± 3 cm/s, which is close to previously published measurements [28,29].

It should also be noted that one of the rat hearts (Fig 8A and 8B) was paced from the atrio-ventricular groove. We can see simultaneous AP propagation in the atria and ventricles from the pacing electrode, which demonstrates that both atrial and ventricular activation sequence can be accurately recorded with a cheaper optical mapping system. The iDS system was also capable of recording atrial AP waveform as seen in Figs 3C and 8B, however because of uneven illumination by excitation LED the recording from right atrium was noisy.

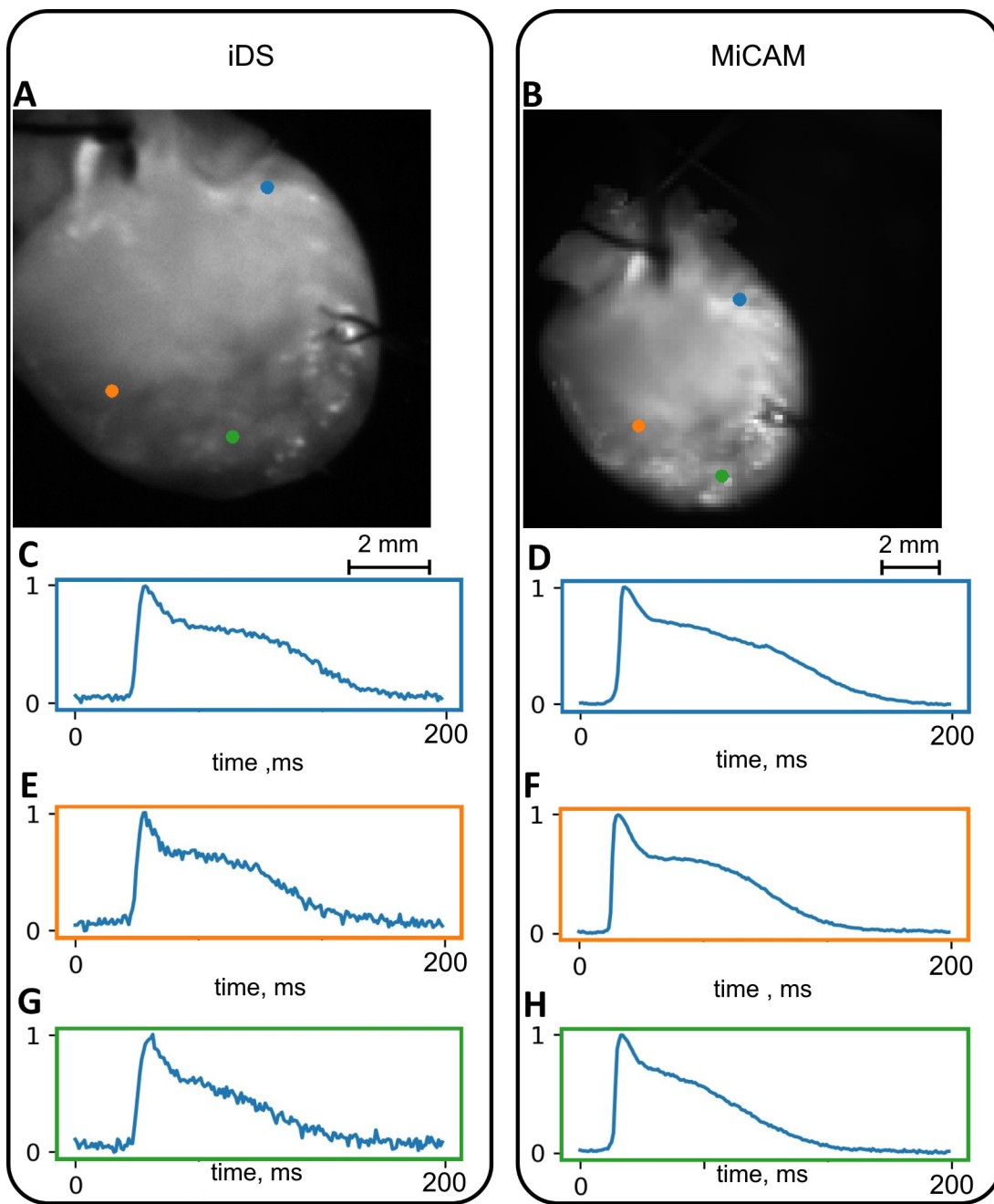

**Fig 3. Conditioned signals in recordings of a mouse heart.** (A),(B) Still frames with color-coded pixels corresponding to the AP waveforms. (C),(E),(G) Signals obtained from iDS camera. (D),(F),(H) Signals obtained from the MiCAM camera.

## Discussion

We have implemented a low-cost open-source optical mapping system, capable of 977 FPS 200x200 pixel imaging. The total system price, as shown in S1 Table, is below US$5,000 including the light source, at the time of publication, while the gold standard Micam Ultima-L system used in this study for comparison is priced at or above US$100,000. It should be noted that the main goal of the study was to test the iDS camera capabilities and limitations. Therefore, we

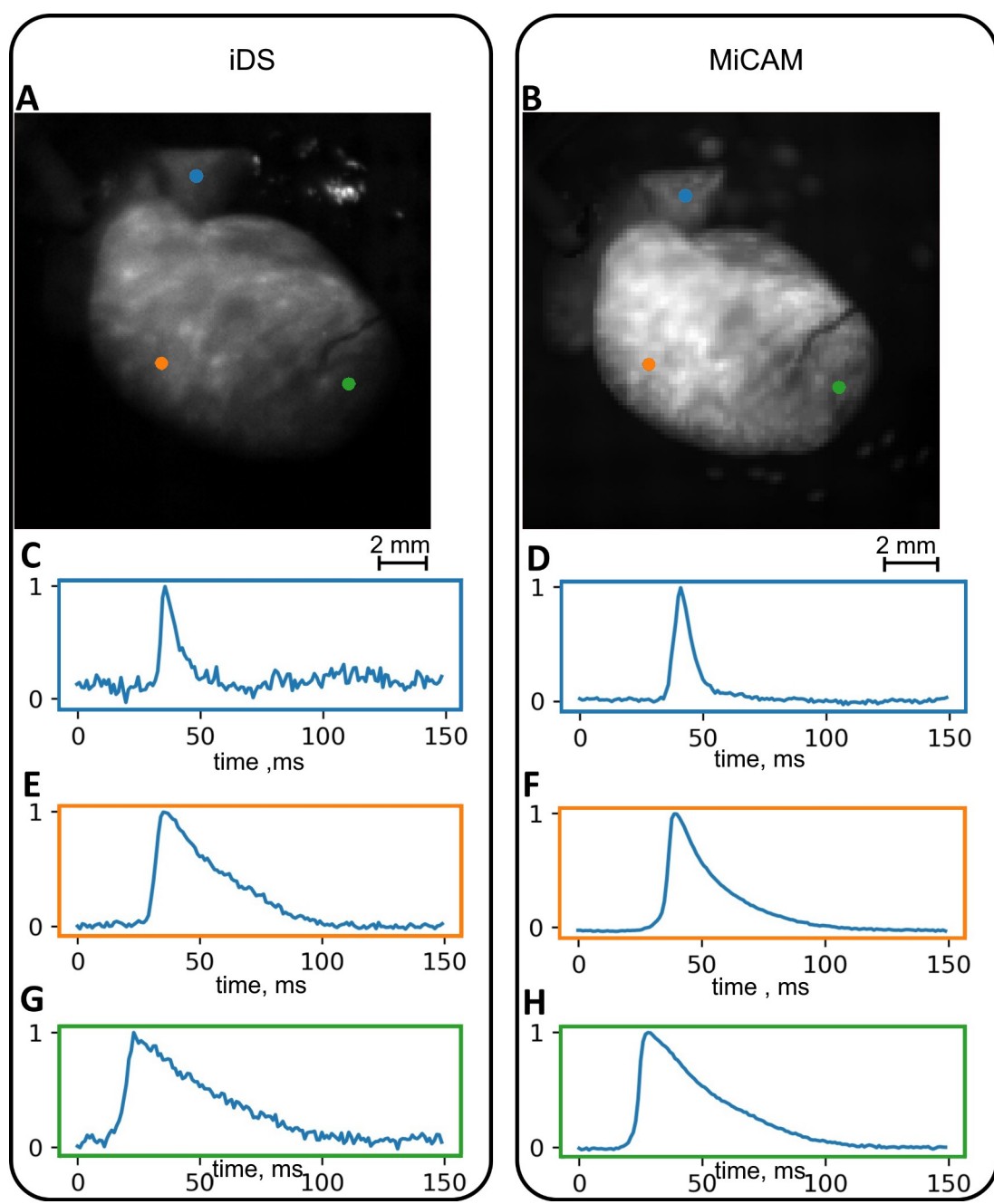

**Fig 4. Conditioned signals in optical recordings of a rat heart.** (A),(B) Still frames with color-coded pixels corresponding to the AP waveforms. (C),(E),(G) Signals obtained from iDS camera. (D),(F),(H) Signals obtained from the MiCAM camera.

kept all components apart from camera and optics equal in both optical mapping system set-ups. As a consequence, the LED we used for die excitation was the most expensive part of the optical mapping system, while the camera and lens themselves were priced below US$1,000.

In our study we have shown that iDS system recordings are of sufficient quality for AP waveform (Figs 3 and 4), activation sequence (Figs 7A and 8A) and CV (Fig 6B) measurements, at 1/20 price. Low signal amplitude resulted in SNR of raw recordings of 0.5±0.4 for the mouse heart and 1.4 ± 0.3 for the rat heart. After signal conditioning, including binning and

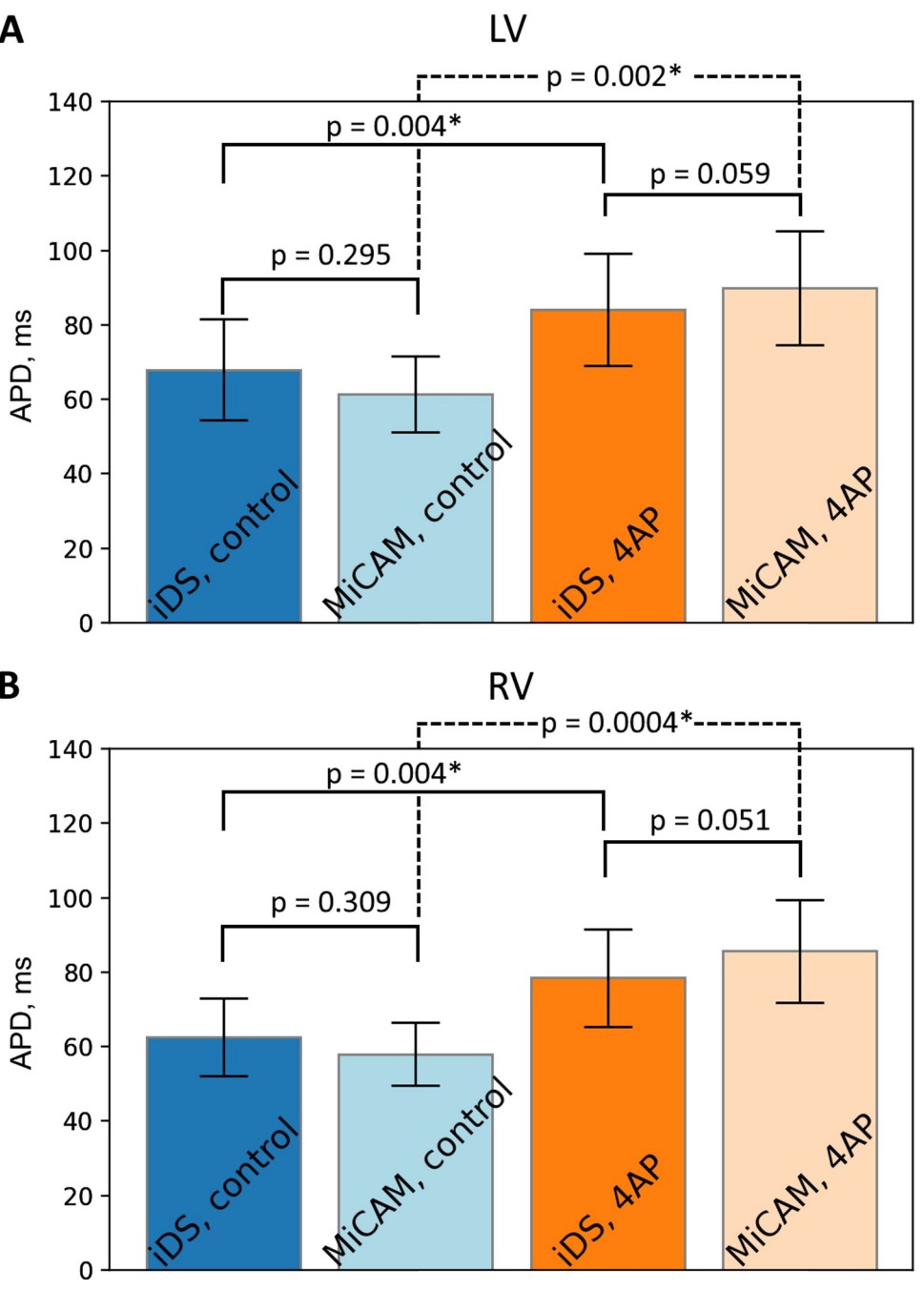

**Fig 5. Comparison of APD measured by iDS and MiCAM.** (A) Comparison of APD in left ventricle. (B) Comparison of APD in right ventricle. Comparison was made for a range of hearts (n = 6) at PCL = 150ms in control and in presence of 5.6mM 4AP. Statistically significant differences are marked by an asterisk (paired t-test).

ensemble averaging, SNR significantly improved to 16±10 for mouse heart and 9±2 for rat heart making it close to MiCAM recordings (Fig 2). As noted above, depolarization phase is less prone to noise then relatively slow repolarization. Consequently, activation sequence and CV measurements by iDS system were robust: RMSD between corresponding recordings by two cameras was equal to 4 cm/s even though ensemble averaging over 2 second recording

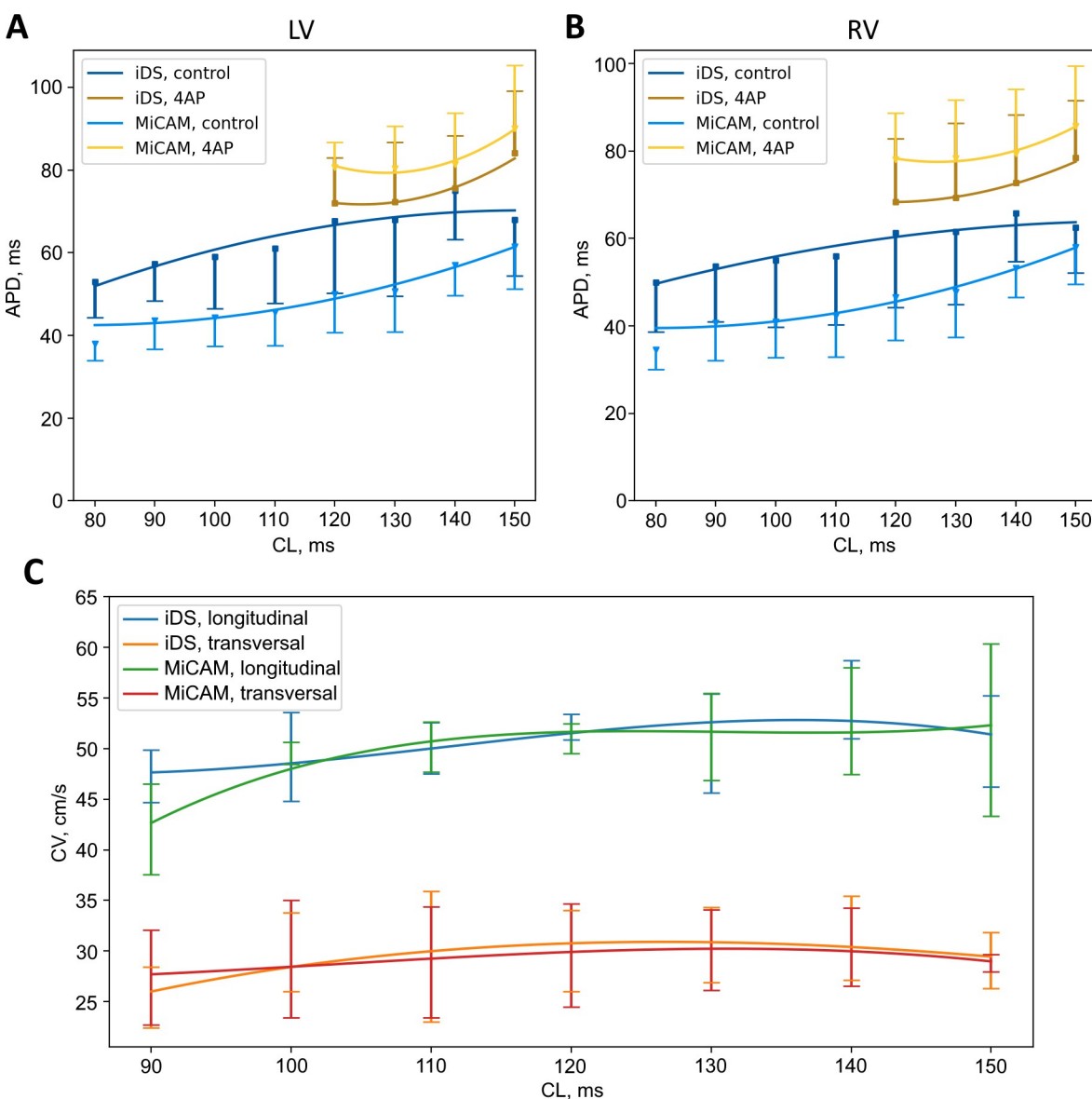

**Fig 6. APD and CV restitution curves.** (A,B) APD80 restitution in control and in presence of 5.6 mM 4AP in left ventricle. (C) Longitudinal/transversal CV restitution measured by iDS and MiCAM cameras.

was not used for these measurements. On the other hand the difference between APD measurements was more pronounced. For example, iDS measurements at 150 ms PCL w/o drug were on average 6 ms longer than MiCAM measurements (Fig 6A). This difference is, in part, due to the distortion of the depolarisation phase and resting potential level by noise. The latter affects the 80% repolarization level hampering the comparison between cameras with different SNR. However, in the worst-case scenario, the difference between APD measurements was 25 ms despite the fact that SNR was high in the iDS recording (SNR equals 25 for this heart, while the average SNR was 16, representative SNR maps for all recordings can be found in S3 Fig). We conclude that largest deviations between the recordings are actually caused by the fact that recordings were not simultaneous, and the field of view was different for the two cameras. This is also consistent with the fact that unpaired comparison between two cameras resulted in high

p-values (unpaired t-test, p = 0.42 in control, p = 0.52 in 4-AP, compare to paired t-test in Fig 5). Despite this difference between the recordings, we have demonstrated that the iDS system is feasible for drug effect measurements. In particular, we have shown significant prolongation of APD by the transient outward current blocker 4-AP (Fig 5, 18±6 ms, n = 5, p = 0.0035, paired t-test). Similar APD prolongation at low drug concentrations was reported previously in isolated mouse ventricular cells [17] and small tissue atrial preparations [30].

Previously Lee et al. had already demonstrated iDS low-cost cameras for panoramic optical mapping [14], using UI-3360CP camera at 1,000 FPS, 160x220 pixels, yielding SNR of about 35 for pig hearts and SNR of about 10 for rabbit hearts at 400 FPS, but the software used in the study was not openly available and the method was not widely disseminated in cardiovascular research community. During the preparation of this publication, Lee et al. published a report of a complete and low-cost optical mapping system, which includes a Langendorff perfusion system complete with pumps and a thermostat with custom controllers, as well as an LED system and an open-source code for low-cost iDS camera [31]. However, this competing solution is not readily compatible with currently available GUI-based signal conditioning and analysis software [9,32–34] and it lacks a graphical user interface suggesting the use of separate *uEye cockpit* application to refocus the camera, adjust the LED intensity and camera gain. The authors presented recordings at 500 FPS, which is often not enough for rodent hearts, in which the entire heart activation takes about 10 ms (Figs 7 and 8). The validation of the system in the study was conducted on large mammals, pig and rabbit hearts, while lacking direct comparison to established optical mapping systems. It remains to be shown that their method is applicable to more popular rodent heart models.

In our work, we focused on the development of optical mapping solution that can be used by biomedical researchers and educators lacking programming or electrical engineering skills. We present a custom open-source software that provides a graphical user interface, convenient interactive camera settings and a real-time viewfinder feed (S1 Fig) and is compatible with established open source Rhythm analysis software that was used through the past decade by many research laboratories [9,32]. The solution including camera and custom open-source software was proven to produce accurate recordings by direct comparison to a specialized optical mapping system. Moreover, our study demonstrates that AP measurements on mouse and rat ventricles (thickness 1.5 and 1.9 mm) and atria are possible, while the previous studies focused on much larger mammals: pig (thickness 20 mm) and rabbit ventricles (thickness 5 mm) [35–38]. While larger hearts yield higher signal intensity, we have demonstrated that at 977 FPS, 200x200 pixels the iDS solution is sufficient not only for AP waveform recordings in small rodent hearts, but also for accurate measurements of both longitudinal and transversal CV in the mouse heart. The high flexibility of software also allows for long recordings. We have tested up to five minutes recordings. It may prove to be essential for the measurement of slower NADH changes during ischemia/reperfusion studies [7,39]. Signal acquisition and processing software used in this study are open-source and distributed under MIT license (the links are provided in the Supporting Information, S1 Text).

We summarize the advantages and limitations of the proposed optical system as follows. (1) The price is more than 20 times cheaper than a specialized camera, making the system suitable for educational purposes or for the applications requiring the use of several cameras such as panoramic and multi-parametric mapping. (2) We have shown that despite the relatively low SNR, AP waveform, activation sequence and CV could be accurately measured after the signal processing *via* compatible open-source Rhythm software. It should be noted, that while only spatial binning was used for the activation sequence and CV, precise measurement of the AP waveform required temporal averaging over 2 second recordings. This requires that the AP waveform is limited to regular periodic heart rhythm while, for example, the recording of alternans is not always feasible. Moreover, arrythmia usually results in complex propagation

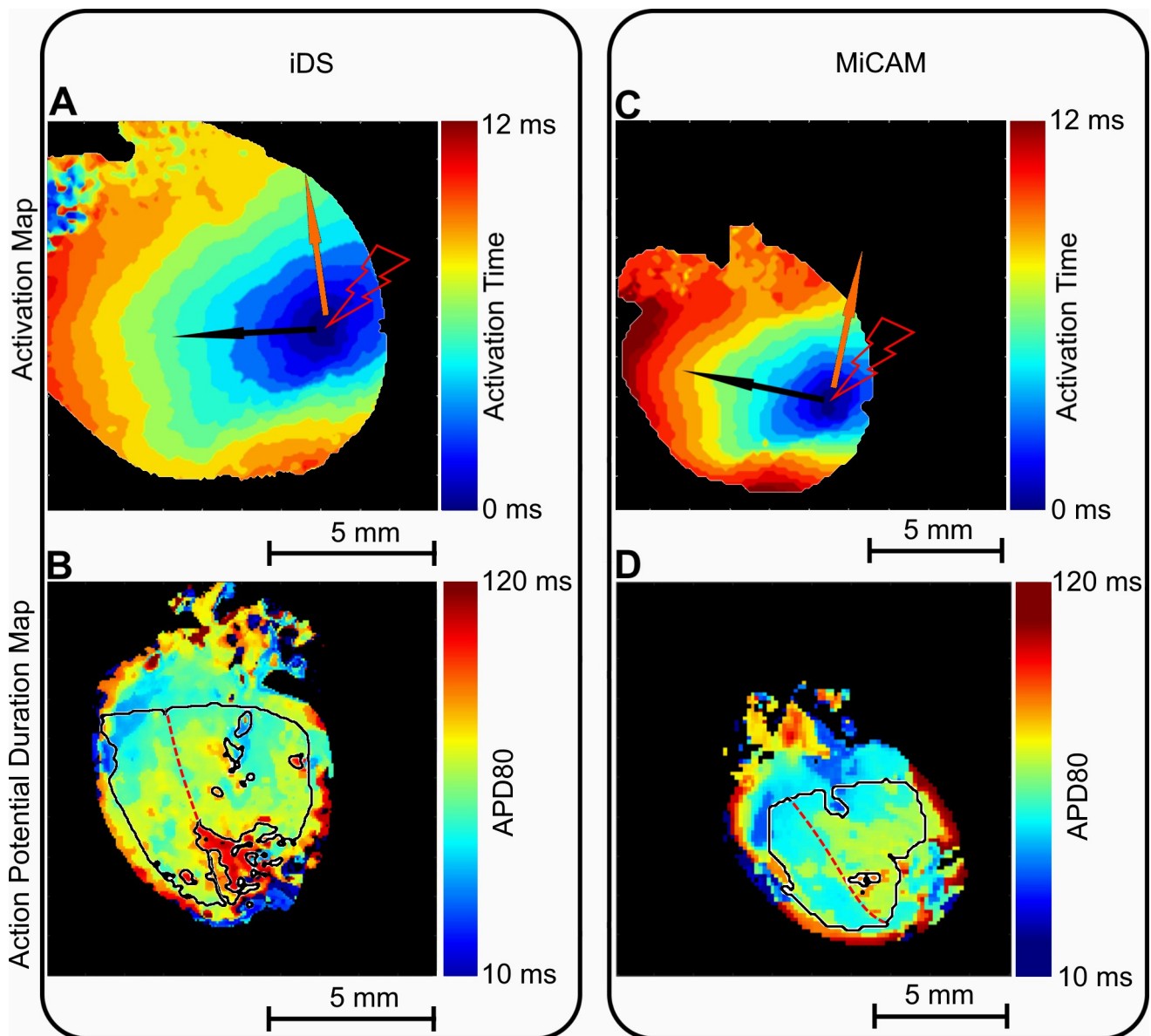

**Fig 7. Comparison of activation maps and APD maps recorded by iDS and MiCAM.** (A) Mouse heart activation sequence measured by iDS camera. (B) Mouse heart APD map measured by iDS camera. (C) Activation sequence measured by MiCAM camera. (D) APD map measured by MiCAM camera. Black line marks the region that was used for APD analysis, the region was divided into left and right ventricle as shown by red dashed line. Pacing electrode location is marked by the lightning symbol, black and orange arrows mark the directions for longitudinal and transversal conduction velocities, correspondingly.

patterns reducing overall SNR of the recording, which is likely to obstruct activation sequence recording as well. (3) As an example of practical application we have demonstrated in this study that drug effects on AP waveform can be measured by the optical-mapping system. However, as exemplified by atrial recordings in Fig 4C, thin myocardium tissue reduces the quality of the signal, which might hinder the use of the cheaper system in experiments involving the mapping of the monolayer cell cultures.

Another limitation is a requirement of a wide-angle lens. UI-3130CP has a 1/3.6" (or 7.1 x 7.1 mm) sensor, while the effective size was reduced to 1x1mm in order to increase the camera

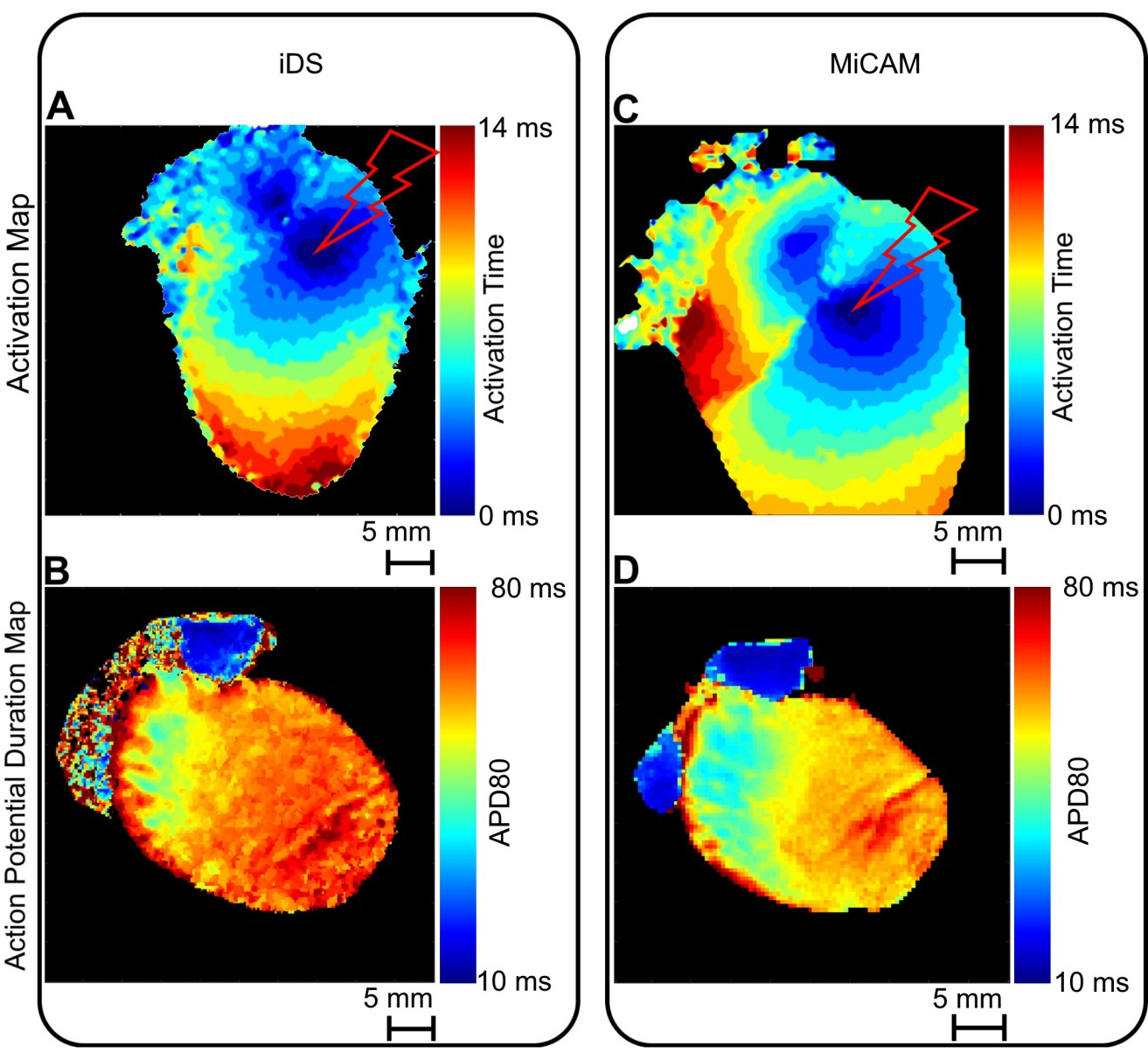

**Fig 8. Comparison of activation maps and APD maps captured by iDS and MiCAM.** (A) Rat heart activation sequence measured by iDS camera. (B) Rat heart APD map measured by iDS camera. (C) Activation sequence measured by MiCAM camera. (D) APD map measured by MiCAM camera. Pacing electrode location is marked by the lightning symbol.

frame rate. Therefore, the lens used in the study (Pentax 6mm TV lens) provided only a 9.2 degrees field of view, which is optimal for the hearts of small rodents such as rats and mice, but a different lens or a tandem optical system should be designed for larger mammals. The use of a more advanced optics could also improve overall signal quality.

## Supporting information

**S1 Fig. Interactive image acquisition software.** Screenshots of image acquisition software. Real-time viewfinder feed allows the user to adjust settings when signal amplitude is too low (B,C) or oversaturating (D).
(TIF)

**S2 Fig. Signal conditioning.** Samples of the signal from a single pixel after different steps of conditioning.
(TIF)

**S3 Fig. SNR and intensity maps.** SNR maps and intensity maps for conditioned recordings of 6 mouse hearts recorded with iDS camera at PCL = 150 ms.
(TIF)

**S1 Table. Component prices in the MiCAM Ultimate-L system and the system presented in this study.**
(DOCX)

**S1 Text. Software links and supporting figures.**
(PDF)

## Author Contributions

**Conceptualization:** Jaclyn Brennan, Igor R. Efimov, Roman Syunyaev.

**Data curation:** Dmitry Rybashlykov, Jaclyn Brennan, Zexu Lin, Igor R. Efimov, Roman Syunyaev.

**Formal analysis:** Dmitry Rybashlykov, Zexu Lin, Roman Syunyaev.

**Funding acquisition:** Igor R. Efimov, Roman Syunyaev.

**Investigation:** Dmitry Rybashlykov, Jaclyn Brennan, Zexu Lin, Igor R. Efimov, Roman Syunyaev.

**Methodology:** Dmitry Rybashlykov, Jaclyn Brennan, Zexu Lin, Igor R. Efimov, Roman Syunyaev.

**Project administration:** Igor R. Efimov, Roman Syunyaev.

**Resources:** Jaclyn Brennan, Zexu Lin, Igor R. Efimov.

**Software:** Dmitry Rybashlykov, Roman Syunyaev.

**Supervision:** Igor R. Efimov, Roman Syunyaev.

**Validation:** Dmitry Rybashlykov, Jaclyn Brennan, Zexu Lin, Igor R. Efimov, Roman Syunyaev.

**Visualization:** Dmitry Rybashlykov, Jaclyn Brennan, Roman Syunyaev.

**Writing – original draft:** Dmitry Rybashlykov, Jaclyn Brennan, Zexu Lin, Igor R. Efimov, Roman Syunyaev.

**Writing – review & editing:** Dmitry Rybashlykov, Jaclyn Brennan, Zexu Lin, Igor R. Efimov, Roman Syunyaev.

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
