## [Decision Letter · Decision Letter 0]

9 Nov 2021

PONE-D-21-32419Open-Source Low-Cost Cardiac Optical Mapping SystemPLOS ONE

Dear Dr. Syunyaev,

Thank you for submitting your manuscript to PLOS ONE. After careful consideration, we feel that it has merit but does not fully meet PLOS ONE’s publication criteria as it currently stands. Therefore, we invite you to submit a revised version of the manuscript that addresses the points raised during the review process.

We look forward to receiving your revised manuscript.

Kind regards,

Alexander V Panfilov, PhD

Academic Editor

PLOS ONE

Journal Requirements:

"NO authors have competing interests."

Additional Editor Comments:

Please respond in full to the comments of the referee and please pay special attention to the comments on clarification of the method, such as re separate figure with all the steps of signal conditioning, for both cameras, including raw signal from one pixel, signals after gaussian filtering and after time averaging. Please also comment if you consider a two-second time averaging as a significant limitation which does not allow mapping of cardiac arrhythmias or unstable spontaneous rhythm where cycle length changes in a beat-to-beat manner. If so, please discuss what should be done in such cases and how it can be addressed/improved.

Reviewers' comments:

Reviewer's Responses to Questions

**Comments to the Author**

1. Is the manuscript technically sound, and do the data support the conclusions?

Reviewer #1: Yes

2. Has the statistical analysis been performed appropriately and rigorously? 

Reviewer #1: Yes

3. Have the authors made all data underlying the findings in their manuscript fully available?

Reviewer #1: Yes

4. Is the manuscript presented in an intelligible fashion and written in standard English?

Reviewer #1: Yes

5. Review Comments to the Author

Reviewer #1: The study by Rybashlykov et al. presents a novel, complete low-cost solution for an optical mapping system. The authors tested an iDS UI-3130CP-M-GL camera-based system and showed that for the most of electrophysiological parameters, including AP duration and restitution, CV and CV anisotropy, as well as pharmacological testing, the system provides reasonable outcomes that are comparable with a 100-times more expensive MiCAM ULTIMA-L system.

The study is timely and innovative; it presents an important step into the development of reliable low-cost optical mapping systems which are specifically important for pharmacological testing. The manuscript is well-written and the results are well-presented. I just have some minor concerns and suggestions on how to improve data presentation and discussion.

First, it would be highly beneficial for the readers if the authors show in a separate figure all the steps of signal conditioning, for both cameras, including raw signal from one pixel, signals after gaussian filtering and after time averaging.

Two-second time averaging sounds like a significant limitation which does not allow mapping of cardiac arrhythmias or unstable spontaneous rhythm where cycle length changes in a beat-to-beat manner.

Figure S2: It appears that there is a high variability of SNR between 7 mouse hearts shown. Two hearts have a very high SNR but in other three hearts SNR is significantly lower. I was wondering if SNR depends on the quality of staining? Can the authors show the SNR images overlapped with the corresponding fluorescence images (similar to those on Fig. 3A, B and Fig. 4A, B)?

Hearts shown in Figure 3B and Figure 4A seem to be out of focus.

Figure 6: APD difference of 9 ms between the systems sounds too high (about 17% of baseline APD). Was this difference statistically significant? Where were these APDs calculated? Were they averaged throughout the entire mapped area of both ventricles? Would it be more accurate to measure APD separately in the right and the left ventricles? This may decrease the observed difference in APD obtained by different systems.

I assume that APD heterogeneity (dispersion) within the area of interest was relatively high that may also affect the average APD values shown and thus contribute to the observed difference. It would be beneficial to show APD heterogeneity measured within the area of interest for each system.

Figure 7: For both systems, APD looks very heterogeneous within the ventricles. Moreover, there are some regions with outliers: APD of 10 ms (dark blue areas) and more than 100 ms (dark red areas) seems to be not physiological. It would be helpful if the authors exclude these outliers and re-analyzed their data for Figure 5 and 6. It may significantly improve the results.

Finally, the authors may consider including a special section in the discussion that would summarize major pros and contras for the presented system. This section should include the mention that this system might be not applicable for specific studies focused on cardiac arrhythmias (see above).

6. PLOS authors have the option to publish the peer review history of their article (what does this mean?). If published, this will include your full peer review and any attached files.

Reviewer #1: No

---

## [Author Response · Author response to Decision Letter 0]

8 Dec 2021

We thank the editor and the reviewer for their questions. We hope that we were able to address the concerns in the new revision. In particular, we have revised the text and the figures and did and additional experiment in order to improve Figures 4,8 quality. We have also added a separate figure with all the steps of signal conditioning for both cameras. The limitations of a two-second time averaging is also discussed in the new version of the manuscript.

The text below is also attached to the submission as a separate file.

"We thank the reviewer for the positive response to our manuscript. We have carefully considered the suggestions, ran an additional experiment and revised the text and the figures. For the convenience, the reviewer’s questions are colored with cyan font below.

Reviewer #1: The study by Rybashlykov et al. presents a novel, complete low-cost solution for an optical mapping system. The authors tested an iDS UI-3130CP-M-GL camera-based system and showed that for the most of electrophysiological parameters, including AP duration and restitution, CV and CV anisotropy, as well as pharmacological testing, the system provides reasonable outcomes that are comparable with a 100-times more expensive MiCAM ULTIMA-L system.

The study is timely and innovative; it presents an important step into the development of reliable low-cost optical mapping systems which are specifically important for pharmacological testing. The manuscript is well-written and the results are well-presented. I just have some minor concerns and suggestions on how to improve data presentation and discussion.

First, it would be highly beneficial for the readers if the authors show in a separate figure all the steps of signal conditioning, for both cameras, including raw signal from one pixel, signals after gaussian filtering and after time averaging.

We have added the S2 figure showing the signal during all steps of signal processing. We thank the reviewer for the suggestion and believe that new figure makes the signal quality of the camera much more clear for a potential reader.

Please see lines 134-140:

“The signals after all intermediate steps of signal conditioning for both cameras are shown in the S2 Figure. Raw iDS-recorded signal was, in general, noisier as compared to signals from MiCAM camera. While spatial filtering reduced the SNR to a level adequate to measure upstroke for activation sequence and conduction velocity (CV) analysis, both spatial and temporal averaging were required to measure APD. Therefore, only the former step of signal conditioning was used for the activation sequence and conduction velocity (CV) measurements.“

Two-second time averaging sounds like a significant limitation which does not allow mapping of cardiac arrhythmias or unstable spontaneous rhythm where cycle length changes in a beat-to-beat manner.

Indeed, this is a significant limitation. It should be noted though that in our publication we didn’t use the time averaging to measure the activation sequences and the conduction velocities. The reason behind that is that the upstroke of a signal is more robust and easier to measure then slower repolarization. Nevertheless the system might not allow the user to accurately measure the activation sequences in case of arrhythmia, because in general the signal tend to be more noisy in this case. We have added the text discussing both the limitations of the optical mapping system and differences between the measurements of APD and activation sequence.

Lines 244-246:

“RMSD between corresponding recordings by two cameras was equal to 4 cm/s even though ensemble averaging over 2 second recording was not used for these measurements ”

Please also see new lines 134-140 quoted above.

Lines 299-315 discussing the limitations are also added and quoted at the very end of this Response document.

Figure S2: It appears that there is a high variability of SNR between 7 mouse hearts shown. Two hearts have a very high SNR but in other three hearts SNR is significantly lower. I was wondering if SNR depends on the quality of staining? Can the authors show the SNR images overlapped with the corresponding fluorescence images (similar to those on Fig. 3A, B and Fig. 4A, B)?

We have added the amplitude of the fluorescence to the figure S2 (our original submitted version) but now the figure S3 (the new version of the manuscript). We believe that although there is some correlation between SNR and signal intensity, some other factors are involved, namely the uneven illumination of heart surface by a LED and overall viability of the tissue. 

Hearts shown in Figure 3B and Figure 4A seem to be out of focus.

Thank you for pointing out this issue, we have changed the figure 3B (mouse heart) and redone the experiment with the rat heart (Fig. 4, and Fig. 8B,D). We believe that the quality of both the heart images and the recordings themselves have improved.

Figure 6: APD difference of 9 ms between the systems sounds too high (about 17% of baseline APD). Was this difference statistically significant? Where were these APDs calculated? Were they averaged throughout the entire mapped area of both ventricles? Would it be more accurate to measure APD separately in the right and the left ventricles? This may decrease the observed difference in APD obtained by different systems. 

I assume that APD heterogeneity (dispersion) within the area of interest was relatively high that may also affect the average APD values shown and thus contribute to the observed difference. It would be beneficial to show APD heterogeneity measured within the area of interest for each system.

Figure 7: For both systems, APD looks very heterogeneous within the ventricles. Moreover, there are some regions with outliers: APD of 10 ms (dark blue areas) and more than 100 ms (dark red areas) seems to be not physiological. It would be helpful if the authors exclude these outliers and re-analyzed their data for Figure 5 and 6. It may significantly improve the results.

Indeed, the difference between the APDs as measured by two cameras is the issue. In a previous version of the manuscript the APD was averaged over the whole mapped area of both ventricles, excluding the regions with high SNR and over-saturating areas. The difference was not statistically significant, but p-value was relatively low (p=0.1). We have redone the analysis separating the ventricles and excluding the APD outliers. While the results improved, but the mean difference is still relatively high: 7+/- 12 ms for LV (p=0.30, paired t-test), 5+/-9 ms for RV (p=0.31). We believe that there are two reasons behind these differences: 1) some changes due to imperfect perfusion happened to the hearts between the measurements by two optical systems and 2) field of view was not exactly the same. 

We thank the reviewer for the suggestion and have modified Figs 5 and 6. We have also outlined the region included in statistical analysis in the new version of Fig. 7 to make it clear for a reader. The numbers throughout the text were changed in accordance with new analysis.

Please also see new lines 149-152:

“APD outlier boundaries were calculated as Q1 - 1.5*IQR and Q3 + 1.5*IQR, where Q1 and Q3 denote first and third quartile respectively and IQR = Q3 - Q1 (interquartile range). Any value exceeding those boundaries was considered an outlier and was excluded from statistical analysis.”

Lines 177-178:

“APD80 measurements in left ventricle by two cameras differed by 7 ± 12 ms in control and 5 ± 5 ms in 4-AP; in right ventricle: 5 ± 9 ms in control, 6 ± 5 ms in 4-AP.”

Lines 256-258:

“This is also consistent with the fact that unpaired comparison between two cameras resulted in high p-values (unpaired t-test, p=0.42 in control, p=0.52 in 4-AP, compare to paired t-test in Fig. 5).“

Finally, the authors may consider including a special section in the discussion that would summarize major pros and contras for the presented system. This section should include the mention that this system might be not applicable for specific studies focused on cardiac arrhythmias (see above).

Thank you we have added the section. Please, see lines 299-315:

“We summarize the advantages and limitations of the proposed optical system as follows. (1) The price is more than 20 times cheaper than a specialized camera, making the system suitable for educational purposes or for the applications requiring the use of several cameras such as panoramic and multi-parametric mapping. (2) We have shown that despite the relatively low SNR, AP waveform, activation sequence and CV could be accurately measured after the signal processing via compatible open-source Rhythm software. It should be noted, that while only spatial binning was used for the activation sequence and CV, precise measurement of the AP waveform required temporal averaging over 2 second recordings. This requires that the AP waveform is limited to regular periodic heart rhythm while, for example, the recording of alternans is not always feasible. Moreover, arrythmia usually results in complex propagation patterns reducing overall SNR of the recording, which is likely to obstruct activation sequence recording as well. (3) As an example of practical application we have demonstrated in this study that drug effects on AP waveform can be measured by the optical-mapping system. However, as exemplified by atrial recordings in Fig. 4 C, thin myocardium tissue reduces the quality of the signal, which might hinder the use of the cheaper system in experiments involving the mapping of the monolayer cell cultures.“"

---

## [Decision Letter · Decision Letter 1]

23 Feb 2022

Open-Source Low-Cost Cardiac Optical Mapping System

PONE-D-21-32419R1

Dear Dr. Syunyaev,

We’re pleased to inform you that your manuscript has been judged scientifically suitable for publication and will be formally accepted for publication once it meets all outstanding technical requirements.

Kind regards,

Alexander V Panfilov, PhD

Section Editor

PLOS ONE

Additional Editor Comments (optional):

Reviewers' comments:

Reviewer's Responses to Questions

**Comments to the Author**

1. If the authors have adequately addressed your comments raised in a previous round of review and you feel that this manuscript is now acceptable for publication, you may indicate that here to bypass the “Comments to the Author” section, enter your conflict of interest statement in the “Confidential to Editor” section, and submit your "Accept" recommendation.

Reviewer #1: All comments have been addressed

2. Is the manuscript technically sound, and do the data support the conclusions?

Reviewer #1: Yes

3. Has the statistical analysis been performed appropriately and rigorously? 

Reviewer #1: Yes

4. Have the authors made all data underlying the findings in their manuscript fully available?

Reviewer #1: Yes

5. Is the manuscript presented in an intelligible fashion and written in standard English?

Reviewer #1: Yes

6. Review Comments to the Author

Reviewer #1: The authors successfully addressed all my questions and concerns. I do not have additional comments.

7. PLOS authors have the option to publish the peer review history of their article (what does this mean?). If published, this will include your full peer review and any attached files.

Reviewer #1: No

---

## [Editor Report · Acceptance letter]

28 Feb 2022

PONE-D-21-32419R1 

Open-source low-cost cardiac optical mapping system 

Dear Dr. Syunyaev:

I'm pleased to inform you that your manuscript has been deemed suitable for publication in PLOS ONE. Congratulations! Your manuscript is now with our production department. 

Kind regards, 

on behalf of

Prof. Alexander V Panfilov 

Section Editor

PLOS ONE